## *Article*

# Optimal Schedule the Operation Policy of a Pumped Energy Storage Plant Case Study Zimapán, México

Gerardo Acuña *[ID], Ramón Domínguez, Maritza L. Arganis and Oscar Fuentes

Engineering Institute, School Circuit, Engineering S/N, C.U., National Autonomous University of Mexico, Coyoacán, Mexico City 04510, Mexico
* Correspondence: gacuna00@gmail.com; Tel.: +52-55-5410-9317

**Abstract:** Pumped-storage hydroelectric plants are an alternative to adapting the energy generation regimen to that of the demand, especially considering that the generation of intermittent clean energy provided by solar and wind power will cause greater differences between these two regimes. In this research, an optimal operation policy is determined through a simulation tool that allows the annual benefits under the energy arbitration service (purchase–sale) to be estimated, considering the variations of the energy price in Mexico. A case study is proposed in the Zimapán hydroelectric facility, where reservoir operation at the hourly level is simulated with records for a period of 3 years, considering historical values. The results establish that this type of pumped storage power plant obtains greater benefits by generating electrical energy during 8 h of high demand and pumping for more than 11 continuous hours in times of low demand. With this configuration, the PHES consumes 82.33 GWh/year more energy than it produces, and the energy generated is 210.83 GWh/year; however, when considering the energy arbitration service, a net income of more than USD 3.25 million per year is identified, which represents a 123.52% increase for the annual energy purchase.

**Keywords:** PHES operation; reservoir operation simulation; simulation tool; operation policy; energy arbitration opportunities; optimal schedule

## 1. Introduction

With the help of PHES systems, energy storage has acquired more importance at the beginning of the 21st century, but its potential use has been identified since the end of the 19th century [1]. Several studies have begun to use simulations on the reservoirs' operation [2], managing to demonstrate the economic benefits of operating their available water resources mathematically [3]; more recently, it was proposed that their operation could be maximized by recirculating the water through the hydraulic regulation tanks [4].

Recently, the procedures to define PHES systems have been based on dynamic simulation models, but none of them has explored using these systems to establish operating policies in an optimal way that benefits their implementation on existing hydroelectric facilities.

In recent years, some countries have increased their investments to transform existing hydroelectric facilities into PHES [5]. However, there are still doubts about the viability of this technology compared to other electricity generation technologies; Abdellatif et al. [6] defined key factors affecting the viability of building a PHES in Egypt, and they showed that, as long as the investment cost of PHES does not exceed 4180 USD/kW and that the costs associated with consuming energy for pumping water are equal to zero; for example, the energy is taken from renewable plants connected to the net and it does not not generate electricity during peak hours, PHES would have greater economic competitiveness over conventional fuel plants. This approach involves looking for mechanisms where the PHES consumes energy in the hours of lower demand or when the value of the energy is almost zero.

In the context of the growing participation of intermittent renewable energies, PHES has re-emerged as an economically and technologically acceptable option to manage electricity production during peak hours and storing wind and solar energy, ensuring its quality and continuity [7]; Kougias et al. [8], mentioned that some of these PHES are under-utilized for energy control and management. Hendena et al. [9] evaluated the convenience of installing a PHES and compared it to expanding the transmission network in its capacity, proving it is more favorable to install a PHES when managing energy.

The above benefits show the importance of implementing PHES in Mexico, which allows for managing intermittent energy and deferring investments in the transmission network.

The Sustainable Development Goals (SDGs) consider that by 2030 the percentage of renewable energies and clean technologies in the energy matrix of countries should increase [10]. Within the framework of this international agreement, several studies have been carried out explaining the advantages of using renewable energies over conventional energies [11], as well as analyzing the importance of improving the environment and achieving energy sustainability [12–15].

In this way, this research is related to the fulfillment of SDG 7 by considering the location of pumped storage plants in existing hydroelectric facilities and optimizing their operation, which generates greater benefits with limited water resources.

Simão et al. [16] conducted a study on hybrid solutions, demonstrating their joint operation with different operating principles and constraints using PHES, wind, and solar technologies. On the other hand, Jacob et al. [17] developed a technical-economic evaluation for PHES in India, demonstrating that the PHES operation can be optimized based on the energy price using the energy arbitration service.

The energy arbitration service represents an opportunity for PHES in Mexico since the variation in demand and the prices recorded throughout the day establish the necessary conditions that allow economic transactions to be carried out for the purchase and sale of energy at a better price convenience [18].

According to the Program for the Development of the National Electric System (PRODESEN), the contribution of renewable energy in Mexico is dominated by hydroelectric production; particularly in the year 2019, when hydroelectricity contributed 7% with 23,602 GWh; for the year 2020, it registered 8% with 26,817 GWh, and in 2021 it represented 11% of the total energy generation of the country with 34,717 GWh [19]. This is a sign of the availability that hydroelectric installations must have to ensure a stored volume to supply more energy.

At the same time, wind power generation contributed 5.20% of the total energy in 2019; in 2020, it totaled 6.21% with 19,702 GWh, and by 2021 it increased to 6.41%, equivalent to 21,074 GWh. For its part, photovoltaic generation for the years 2019, 2020 and 2021 registered an increase of 3.10%, 4.99% and 6.15% of electrical energy in the country, respectively [19]. This is a clear trend in how renewable energies are advancing within the energy matrix, but due to their intermittence, they require the support of hydroelectric production.

The experience in the operation of electrical systems has shown that the high penetration of intermittent technologies causes negative effects on frequency regulation, its quality, reserve margins and the useful life of conventional power plants that must cover the demand that is not supplied by the intermittent technology [20].

For example, solar generation during the day reduces the demand for electricity supplied by conventional plants, mainly based on fossil fuels; this favors the reduction of greenhouse gases due to the displacement of this type of plant, coupled with their low production cost [21].

However, the amount of solar generation begins to decrease exactly when the electricity demand enters its peak hours in the afternoon; this causes the start-ups of the plants in reserve [20]; likewise, other technologies must show their availability to cover the demand that solar does not deliver. In Mexico, large hydroelectric and gas-based plants are used to

cover the missing demand. It is important to note that the latter type generates a greater emission of greenhouse gases due to the increase in their contribution [19].

This generates additional pressure on hydroelectric plants, which are responsible for fulfilling demand peaks in short periods of time, are also required to offer high power, and must always be available to provide energy, mainly at sunset [22]. Some examples of dams with energy stored within their reservoirs are Aguamilpa, La Yesca, El Cajón, belonging to the Santiago and Angostura, and Chicoasén and Malpaso of the Grijalva system.

Figure 1 shows the interaction of hydroelectric plants with intermittent renewable energy; photovoltaic and wind technologies were grouped for the period 2016–2021. The graph shows that as of 2019, the hydroelectric facilities modified their generation schedule to support intermittent plants, mainly during high-demand hours.

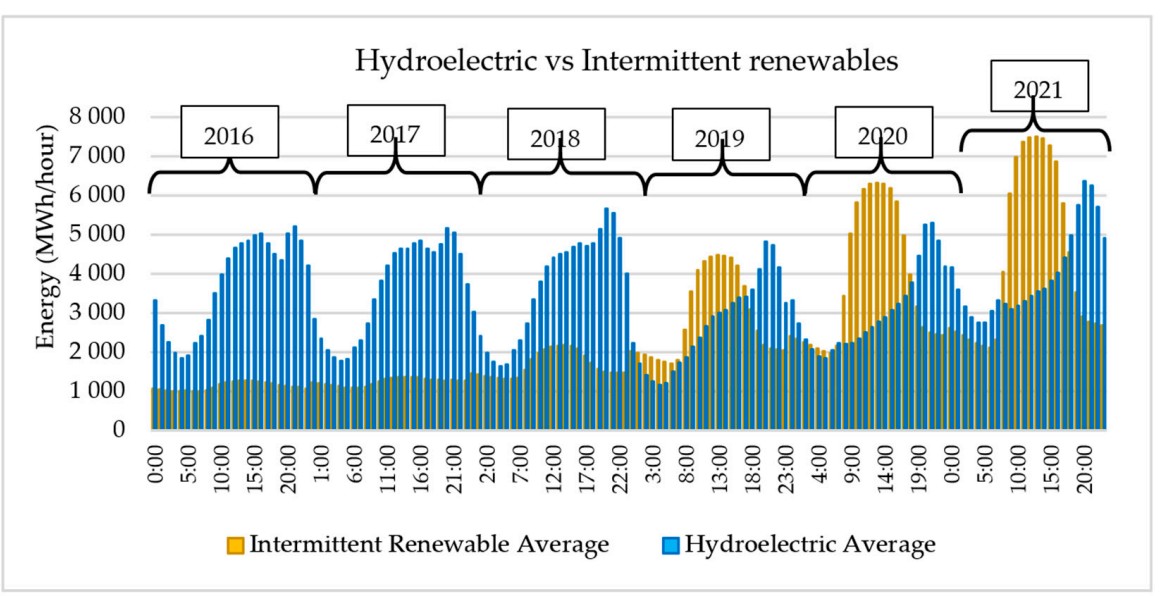

**Figure 1.** Historical participation of hydroelectric plants with variable renewables. Own source with CENACE data, 2021.

In accordance with the renewable energy objectives committed by the country framed in the Energy Transition Law, the goal of 40% of the total energy generation must be produced with renewable energy by the year 2033 and 50% by 2050 [23].

The contribution of renewable plants participating in the three long-term energy auctions launched by private initiative (2015–2017) adds up to a total of 10,646 MW of power to be installed by 2024 with about 30,000 GWh/year [24].

Additionally, strong growth in the installation of small hydroelectrical power plants under the distributed generation scheme is reported. At the end of 2020, there was already an accumulated capacity of 1388 MW, and it is expected to reach 9179 MW by 2035, contributing more than 16,000 GWh/year [25].

This new intermittent energy capacity, intended to be installed in the country, will put the conventional operation of the electricity grid to the test. For this reason, energy storage is a viable solution to address these issues as Mexico moves down the path of energy transition.

As mentioned, hydroelectric facilities are responsible for supplying energy mainly during peak hours, and this situation makes them vulnerable since they must guarantee the availability of water resources when the electrical system requires them. In addition, changing their operation causes early damage to their infrastructure, representing costs in their operation and maintenance that were not considered in their original design [26].

According to PRODESEN 2021, an additional hydroelectric capacity of 1426 MW is projected for the next 15 years by 2036; while its growth for 2025 is just 2.74%; added to

this, the issue of droughts and water scarcity [27] are a latent problem in Mexico, further reducing this projection in its deployment. This low boost to hydroelectric installations in the face of the high growth of renewable plants would complicate the management and control of energy due to the intermittency in the Mexican electricity system.

In Mexico, there are no pumped-storage hydroelectric plants; therefore, to implement an ambitious program of PHES plants, it is necessary to develop well-founded studies in terms of the benefit-cost relationship based on adequate operating policies, the aspect that constitutes the main objective of this research.

Energy storage through PHES can contribute to a greater generation of clean energy at a relatively low cost if they are located on existing hydroelectric plants, taking advantage of the same water resource that enters the reservoir.

The National Energy Control Center always establishes the price of energy, considering the relationship between the supply and demand of the Mexican electricity market, in such a way that when the demand is high (generally between 14 to 22 h), the price is high, while when the demand is lower the price decreases (between 1 to 6 h). This relationship acts on averages, as shown in Figure 2, where it is shown that the price obeys the demand signal, although a certain gap corresponding to the 2018–2020 period is denoted for this case study.

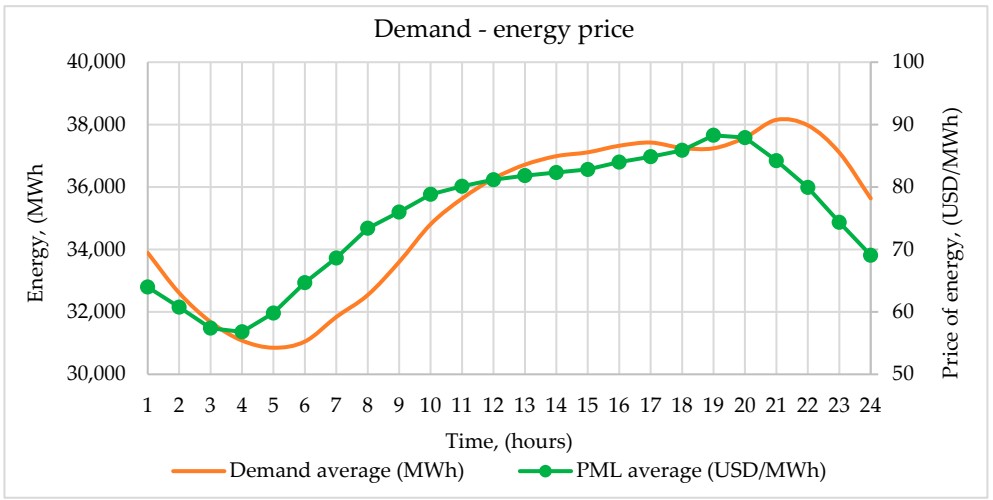

**Figure 2.** Average comparison between demand and energy price. Own source with CENACE data, 2021.

The energy arbitration service is carried out by the National Energy Control Center, which defines hourly prices based on the differences between supply and demand; these prices are updated daily. The operation of a PHES takes advantage of price differences over a period (net income is the difference between the sale and purchase of energy). Figure 3 shows the scheme of the energy arbitration service, where the energy used for pumping in low-demand hours is mainly purchased to be stored through a device that allows the energy to be delivered again in peak or high-demand hours, looking for income from this sale of energy at times of best convenience.

The importance of storage systems in energy generation, in this case, PHES, is that they have the ability to take advantage of the energy at a more valuable time than when it was produced. This service is valuable mainly for integrating intermittent renewable energy into the network [28] since these cannot be programmed to satisfy the demand because their production depends on external and uncontrollable factors such as weather conditions.

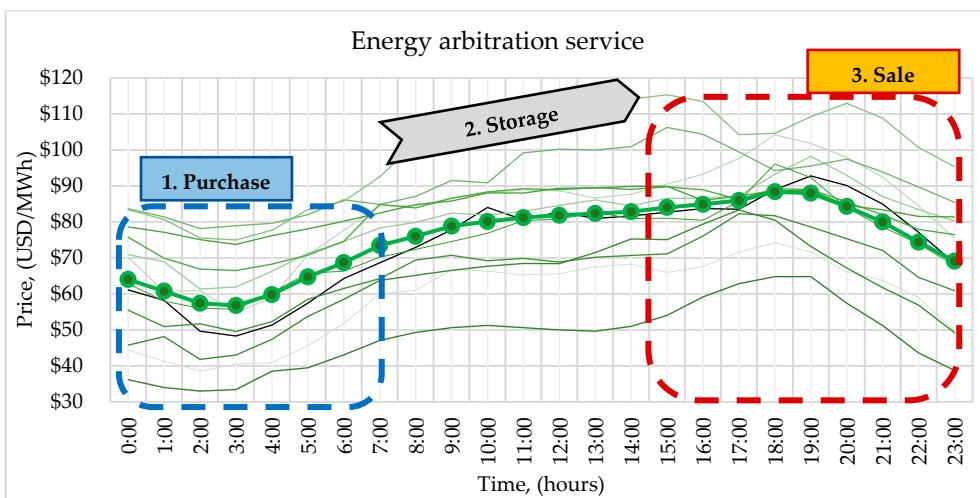

**Figure 3.** Scheme of energy arbitration service. Own source with CENACE data, 2021.

So, by implementing this type of storage solution, it is possible to manage or control the energy that could be saturating the network at inopportune moments; by offering this service to the electricity network operator. The aims are to help the continuity and reliability of the electricity system with variations in supply and demand.

The energy demand can fluctuate from fractions of seconds to daily or weekly periods, even presenting seasonal variations or behaviors. At present, the only storage technology capable of providing this service with large variations over time is PHES, due to its great capacity to store energy in the form of potential energy by connecting hydraulic tanks at different elevations.

The main objective of this paper is to optimize the operation policy, that is, the hours of pumping and generation depending on the energy arbitration service, with the purpose of identifying possible economic benefits of a PHES located on the Zimapán hydroelectric facility as a case study.

In summary:

- It is necessary to increase renewable energies rapidly (Energy Transition Law); of these, wind and solar are the ones that have grown the most and already have proposals. Their problem is that they are intermittent, so at certain times they saturate the system, and at other times they do not help cover demand, so they must be backed up with stored energy that can be made available very quickly.
- Currently, the support is given by hydroelectric plants, but there are already problems because they are made to work with a regime for which they are not designed, and that problem will increase when a greater intermittent energy supply is available.
- According to the above, the use of PHES is attractive; ideally, they take advantage of current conventional hydroelectric plants' infrastructure and storage (of water and energy).
- PHES can benefit financially when schedules are adjusted to take advantage of the hourly differences in the price of energy, which, in turn, depend on the differential between demand and supply.
- It is then necessary to design the operation policy of PHES to achieve the maximum net benefit.

The following conceptual diagram in Figure 4 schematically explains the steps to establish the operation policy of a PHES associated with the local marginal price of energy.

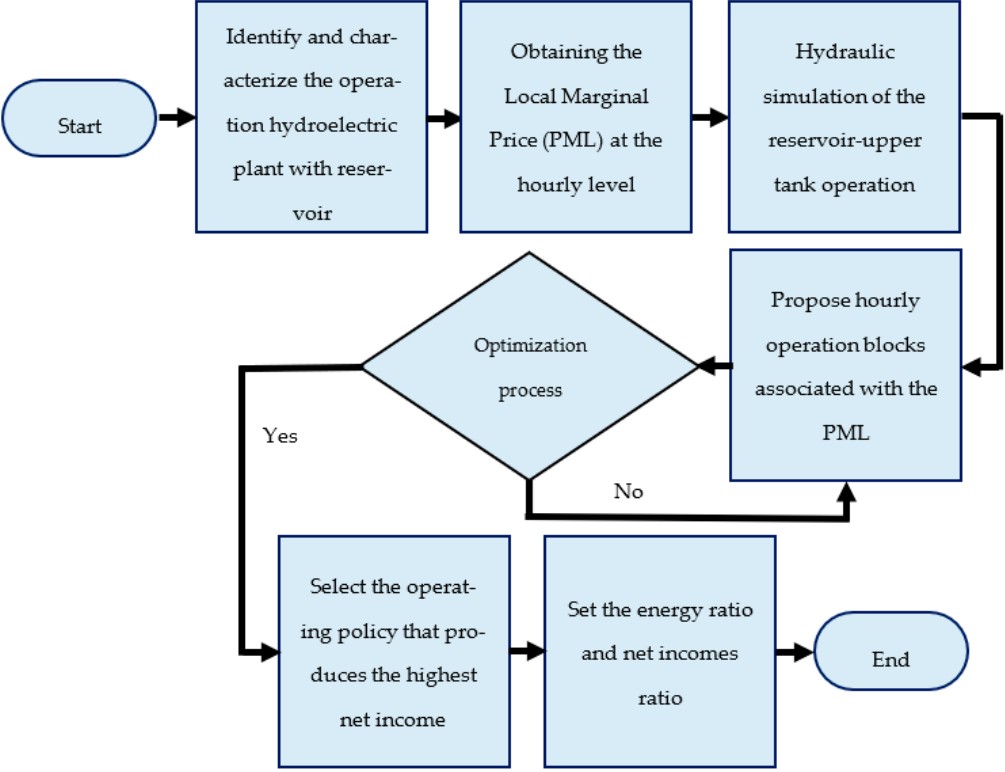

**Figure 4.** Conceptual diagram to establish an optimal operation policy for the PHES.

## 2. Materials and Methods

### 2.1. PHES Operation

A pumped-storage hydroelectric plant is a large-scale storage technology that works by transforming electrical energy into potential energy. For this, two hydraulic regulation tanks are required at different elevations where a certain volume of water is recirculated through a pressure pipe. Firstly, the volume of water is raised to the upper tank through pumping equipment, mainly during hours of low demand. Then, for the high demand hours, the water falls by gravity to the turbines that generate energy, depositing the water again in the lower tank. In this way, the cycle is completed by PHES.

In the case of this study at PHES Zimapán, it is proposed to house the lower hydraulic tank within the reservoir of the Zimapán hydroelectric plant. This allows us to take advantage of the input volumes to carry out the first filling.

This hydroelectric power station was selected as a case study to test the simulation model. Additionally, the Federal Electricity Commission (CFE), which operates the hydroelectric plant, shared public information about three years of operation at the hourly level.

This model could be replicated for all those hydroelectric power stations in the country that have contemplated the installation of PHES in their reservoirs. For circumstances caused by extreme hydro-meteorological events, the PHES station would include spills in its regulation tanks. Therefore, it was also considered that all those spills should be reintegrated into the reservoir to keep the levels and volumes stable with which the hydroelectric facility conventionally operates at the end of the day.

### 2.2. Preliminary Capacity Sizing

To size the lower and upper regulation hydraulic tanks, 0.318% of the useful volume of the Zimapán dam is taken into account, corresponding to 2.26 hm$^3$. This data was the result of a previous analysis to determine the PHES capacity for renewable generation [29], proposing a power density of 10.50 W/m$^2$ with which an area of 7.54 ha and a depth of 30 m is required; the capacity to be installed for PHES was 80 MW, thus complying with the renewable generation index.

With the help of Google Earth© and ARCGIS© software, it was determined that the location of the upper tank would be at a height difference of 420 m, connecting with the reservoir or lower tank through a steel pipe with a length of 700 m and a diameter of 2.45 m. Table 1 summarizes the values of the preliminary size of the PHES Zimapán. Figure 5 schematically shows the location of the PHES Zimapán, where the upper tank and the steel pipe that connects to the lower tank are located.

**Table 1.** Sizing of the PHES Zimapán.

| | | PHES Zimapán | | | |
|---|---|---|---|---|---|
| **Capacity (MW)** | **Height Difference (m)** | **Steel Pipe** | | **Upper Tank** | |
| | | **Diameter (m)** | **Length (m)** | **Volume (hm$^3$)** | **Area (ha)** |
| 80 | 420 | 2.45 | 700 | 2.26 | 7.54 |

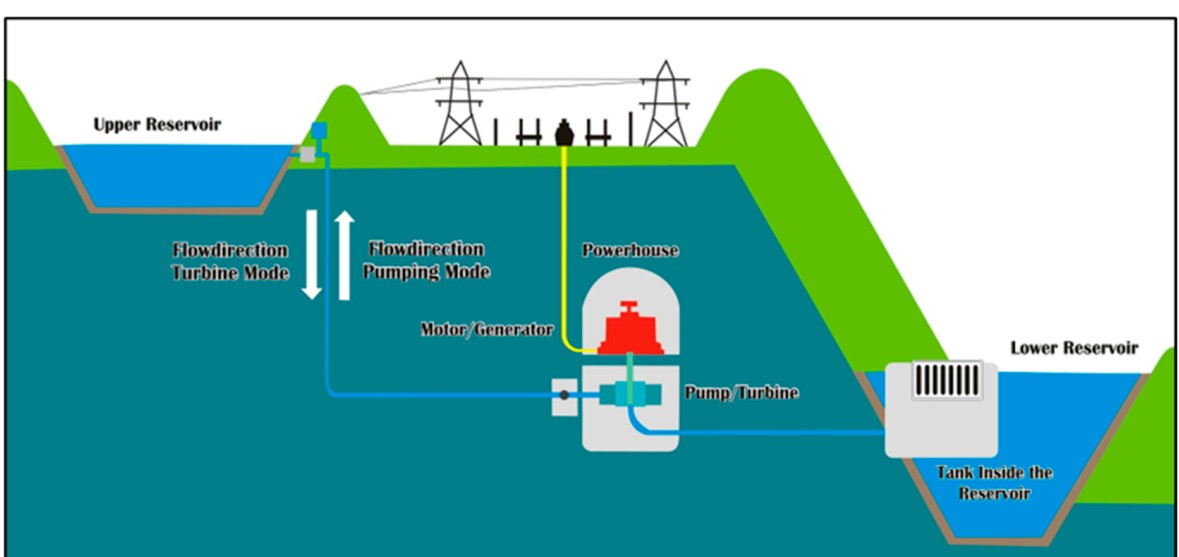

**Figure 5.** Siting scheme of the PHES Zimapán. Own source with IHA information.

Derived from this preliminary sizing, the capacity area elevation curves were defined for both hydraulic tanks, where the minimum water level (mWL) corresponds to the minimum level necessary to avoid the cavitation phenomenon, while the maximum operating water level (MOWL) is the maximum capacity level. If this level is exceeded, the PHES (Tables 1–4) has to spill the additional volume.

**Table 2.** Siting of the PHES Zimapán.

| Siting of the PHES Zimapán | | |
|---|---|---|
| Lower tank base level | 1520 | masl |
| Upper tank base level | 1940 | masl |
| Height difference | 420 | m |
| Steel pipe length | 700 | m |

**Table 3.** Lower Tank Elevations-Capacities-Areas Curve.

| **Lower Tank** | **Elevation (masl)** | **Area (km$^2$)** | **Cap (hm$^3$)** |
|---|---|---|---|
| Base level | 1520 | 0.08 | 0.00 |
| mWL | 1528 | 0.08 | 0.61 |
| MOWL | 1550 | 0.08 | 2.28 |

**Table 4.** Upper Tank Elevations-Capacities-Areas Curve.

| Upper Tank | Elevation (masl) | Area (km$^2$) | Cap(hm$^3$) |
|---|---|---|---|
| Base level | 1940 | 0.08 | 0.00 |
| mWL | 1945 | 0.08 | 0.38 |
| MOWL | 1970 | 0.08 | 2.28 |

### 2.3. Data Set for Analysis

The available public information for the case study was collected and analyzed, characterizing the operating conditions of the Zimapán hydroelectric facility, which has a current capacity of 292 MW and a useful volume of 710 hm$^3$. The hydrological information provided by the Federal Electricity Commission (CFE) consists of three years of operation from January 2018 to December 2020, as described below.

#### 2.3.1. Inlet Volumes and Levels in the Reservoir

According to the historical series of the input volumes associated with the level in the reservoir, it was possible to characterize the operation of the Zimapán hydroelectric plant, defining the input volumes with a daily average of 25.34 hm$^3$. In the case of the water levels in the reservoir, these were recorded below the maximum operating water level (MOWL = 1560 masl) and above the minimum operating water level (mWL = 1520 masl) for the three years, as can be seen in Table 5 and Figures 6 and 7.

**Table 5.** Input daily volumes to the dam reservoir (hm$^3$ per day).

| Input Volume (hm$^3$) | 2018 | 2019 | 2020 | Historical |
|---|---|---|---|---|
| Average | 39.31 | 15.02 | 21.68 | 25.34 |
| Minimum | −0.46 | −30.15 | −20.73 | −30.15 |
| Maximum | 137.16 | 111.17 | 61.92 | 137.16 |
| Standard deviation | 27.50 | 22.22 | 7.83 | 23.26 |

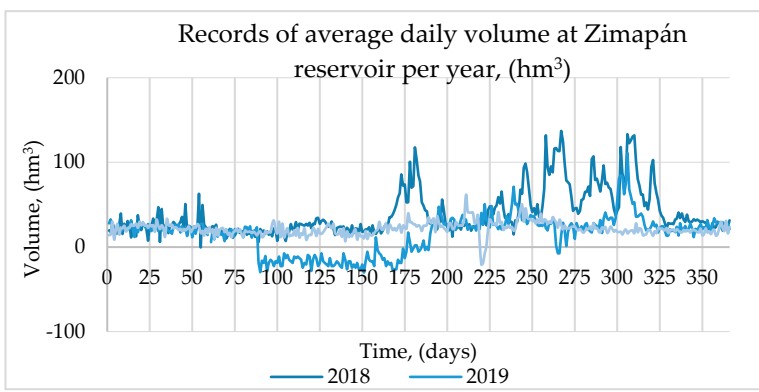

**Figure 6.** Daily average flow records to the Zimapán reservoir. Own source with hydrological information CFE, 2020.

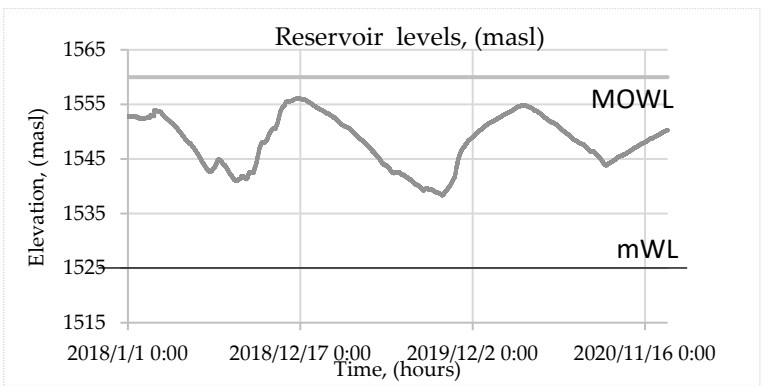

**Figure 7.** Historical levels in the reservoir (masl). Own source with hydrological information CFE, 2020.

2.3.2. Evaporations

To estimate the volume losses in the upper hydraulic tank of the PHES Zimapán, daily information was collected from the LAS ADJUNTAS-HGO station, and information was taken from the national climatological database [30]. Table 6 shows the summary of the information processed for three complete years from the period 1993 to 1995 that were available. To stress the model at maximum values due to evaporation, the maximum monthly value of evaporation (mm) was obtained. These values were assumed for the simulation since no further information was found for the current period.

**Table 6.** Complete monthly evaporation records for the Zimapán reservoir, maximum evaporation sheet (mm).

| Year | Jan. | Feb. | Mar. | Apr. | May. | Jun. | Jul. | Aug. | Sep. | Oct. | Nov. | Dec. | Annual |
|---|---|---|---|---|---|---|---|---|---|---|---|---|---|
| 1993 | 141.7 | 178.9 | 239.2 | 233.2 | 232.7 | 184.7 | 215.3 | 214.2 | 143.7 | 140.4 | 116.1 | 117.0 | 2157.0 |
| 1994 | 106.2 | 140.6 | 204.7 | 201.0 | 211.1 | 184.4 | 219.4 | 174.5 | 160.8 | 143.9 | 139.8 | 136.7 | 2023.1 |
| 1995 | 130.9 | 158.3 | 226.5 | 242.8 | 253.6 | 209.3 | 172.1 | 154.7 | 163.6 | 178.5 | 105.2 | 112.8 | 2108.3 |
| average | 126.3 | 159.3 | 223.5 | 225.7 | 232.4 | 192.8 | 202.3 | 181.1 | 156.0 | 154.2 | 120.4 | 122.2 | 2096.1 |
| Max | 141.7 | 178.9 | 239.2 | 242.8 | 253.6 | 209.3 | 219.4 | 214.2 | 163.6 | 178.5 | 139.8 | 136.7 | 2157.0 |

2.3.3. Local Marginal Price

The local marginal price (PML) reflects the value of energy at a given time and place. Currently, in Mexico, there are approximately 2500 price nodes contained in 108 charging zones and nine regional control departments. Each of the nodes has a price determined by three components: energy, losses, and congestion. The Zimapán hydroelectric, in the state of Querétaro, belongs to node 03ZMN-115; according to CENACE, it is in the Ixmiquilpan loading zone within the Western region [31].

The operation policy allows the association of the local marginal price at the hourly level, identifying the energy price that corresponds to the moment of pumping and the moment of generating energy; at the end of the day, the transactions are accumulated for each period of operation. This way, the optimization will look for the hours where the net income is the maximum when buying and selling electricity.

To take advantage of the signals of the electricity market, where the differences between registered maximum and minimum historical prices are very high, see Figure 8. Therefore, a simulation is proposed at the hourly level taking the registered historical values.

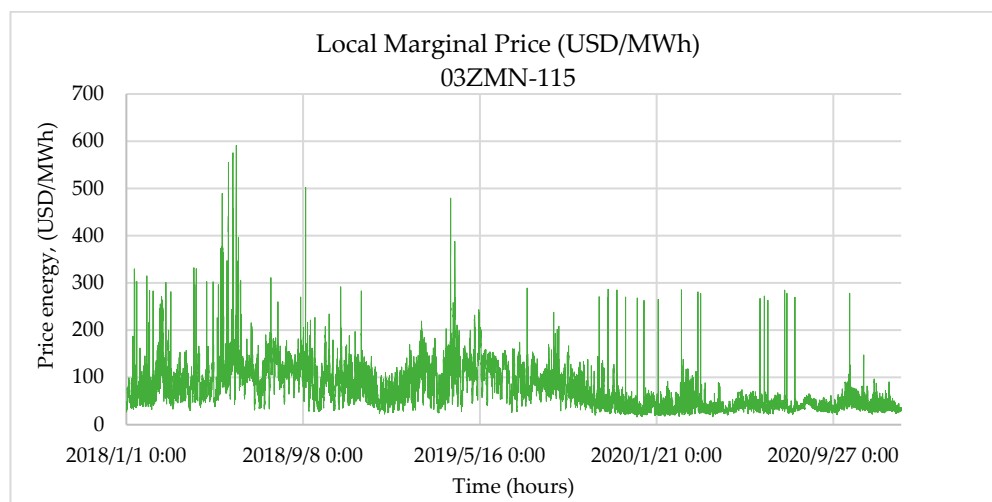

**Figure 8.** Behavior of the price of energy period 2018–2020. Own source with CENACE information, 2020.

Table 7 shows the maximum, minimum and average local marginal price per hour for the period 2018 to 2020. It is divided into spring-summer and autumn-winter seasons in order to appreciate seasonality and its variation at the demand level, which, as explained, is intrinsic in the PML values; that is, the higher the PML price, the higher the energy demand at that hour. Figure 9 shows the historical behavior during the 3 years analyzed.

**Table 7.** Parameters of the local marginal price node 03ZMN-115.

| | Spring (20 March)–Summer (23 September) | | | | Autumn (24 September)–Winter (19 March) | | |
| --- | --- | --- | --- | --- | --- | --- | --- |
| **Hours** | **Min (USD/MWh)** | **Average (USD/MWh)** | **Max (USD/MWh)** | **Hours** | **Min (USD/MWh)** | **Average (USD/MWh)** | **Max (USD/MWh)** |
| **12 a.m.** | 21.77 | 75.88 | 288.73 | 12 a.m. | 17.86 | 46.65 | 233.73 |
| **1 a.m.** | 21.66 | 72.48 | 330.18 | 01 a.m. | 18.48 | 44.12 | 284.69 |
| **2 a.m.** | 21.69 | 68.61 | 183.72 | 02 a.m. | 18.04 | 41.81 | 277.82 |
| **3 a.m.** | 20.67 | 68.20 | 187.70 | 03 a.m. | 17.04 | 42.59 | 263.04 |
| **4 a.m.** | 19.96 | 70.93 | 195.91 | 04 a.m. | 17.68 | 46.40 | 269.58 |
| **5 a.m.** | 18.96 | 74.78 | 195.55 | 05 a.m. | 16.51 | 52.10 | 161.70 |
| **6 a.m.** | 19.68 | 77.56 | 267.15 | 06 a.m. | 16.40 | 56.87 | 162.70 |
| **7 a.m.** | 18.83 | 81.99 | 295.75 | 07 a.m. | 16.46 | 61.03 | 285.44 |
| **8 a.m.** | 16.67 | 85.07 | 376.77 | 08 a.m. | 19.35 | 63.17 | 291.47 |
| **9 a.m.** | 19.23 | 88.16 | 376.38 | 09 a.m. | 18.50 | 65.15 | 275.36 |
| **10 a.m.** | 18.88 | 90.01 | 376.61 | 10 a.m. | 18.35 | 65.34 | 217.42 |
| **11 a.m.** | 18.48 | 92.16 | 440.24 | 11 a.m. | 18.28 | 65.30 | 212.70 |
| **12 p.m.** | 18.94 | 93.07 | 457.25 | 12 p.m. | 17.79 | 65.63 | 280.70 |
| **1 p.m.** | 19.16 | 93.77 | 430.83 | 01 p.m. | 17.78 | 65.75 | 282.46 |
| **2 p.m.** | 20.07 | 93.31 | 504.74 | 02 p.m. | 18.13 | 67.43 | 286.30 |
| **3 p.m.** | 19.47 | 95.09 | 520.08 | 03 p.m. | 19.23 | 68.16 | 204.71 |
| **4 p.m.** | 19.54 | 93.59 | 590.59 | 04 p.m. | 20.83 | 72.16 | 200.76 |
| **5 p.m.** | 19.57 | 91.63 | 457.69 | 05 p.m. | 22.23 | 77.59 | 246.73 |
| **6 p.m.** | 24.02 | 94.28 | 501.99 | 06 p.m. | 21.99 | 79.41 | 260.15 |
| **7 p.m.** | 24.29 | 95.83 | 373.92 | 07 p.m. | 23.44 | 75.92 | 267.60 |
| **8 p.m.** | 23.73 | 93.46 | 552.14 | 08 p.m. | 20.88 | 69.87 | 203.66 |
| **9 p.m.** | 23.91 | 89.60 | 575.49 | 09 p.m. | 20.88 | 64.43 | 270.40 |
| **10 p.m.** | 22.53 | 84.90 | 522.11 | 10 p.m. | 19.01 | 57.72 | 191.98 |
| **11 p.m.** | 21.69 | 81.34 | 332.75 | 11 p.m. | 18.12 | 51.63 | 220.84 |
| **Average** | 20.56 | 85.24 | 388.93 | Average | 18.89 | 61.09 | 243.83 |
| **Minimum** | 16.67 | 68.20 | 183.72 | Minimum | 16.40 | 41.81 | 161.70 |
| **Maximum** | 24.29 | 95.83 | 590.59 | Maximum | 23.44 | 79.41 | 291.47 |
| **Standard deviation** | 2.03 | 9.29 | 127.71 | Standard deviation | 1.89 | 11.14 | 40.84 |

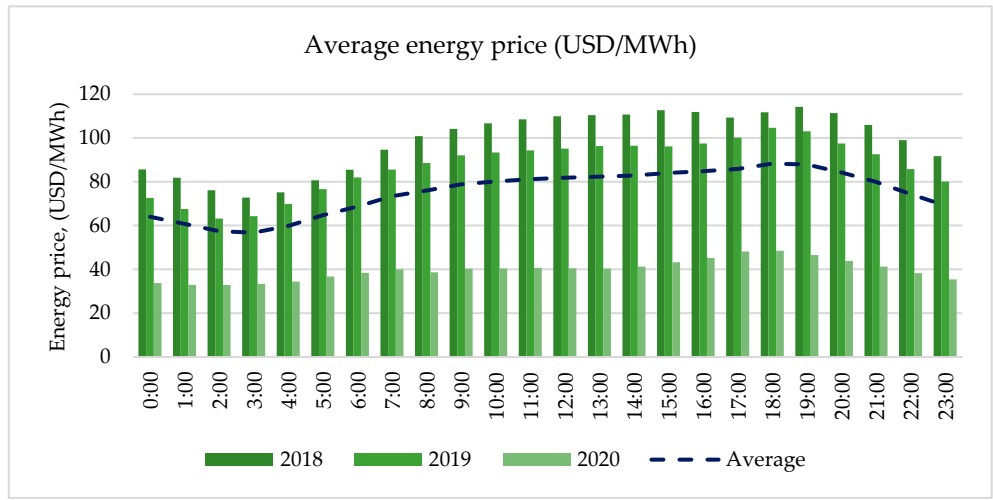

**Figure 9.** Hourly average of the PML per year. Own source with CENACE information, 2020.

Figure 9 plots the hourly average of the PML per year, observing the effects of the COVID-19 pandemic, where energy prices fell by an average of 45.6% in 2020 compared to 2019 due to the economic stagnation of the country.

### 2.4. Optimization Process

Considering the preliminary sizing of the PHES, an operation of the reservoirs is simulated, and the operation policy is optimized for different scenarios, where each of the generation blocks proposed that consider the price of energy is presented every hour, with the aim of finding the right moment for the purchase and sale of energy. The platform used to carry out the simulation of the operation of reservoirs and its subsequent optimization is Microsoft Excel and the implementation of Macros.

To simulate the scenarios, first, the quantity of the water resource and its variability during the period of 2018–2020 are evaluated. Likewise, the evaporation on the upper tank is determined and the hourly prices of energy are considered in the same period of analysis.

The following flowchart (Figure 10) shows the methodology applied to the case study:

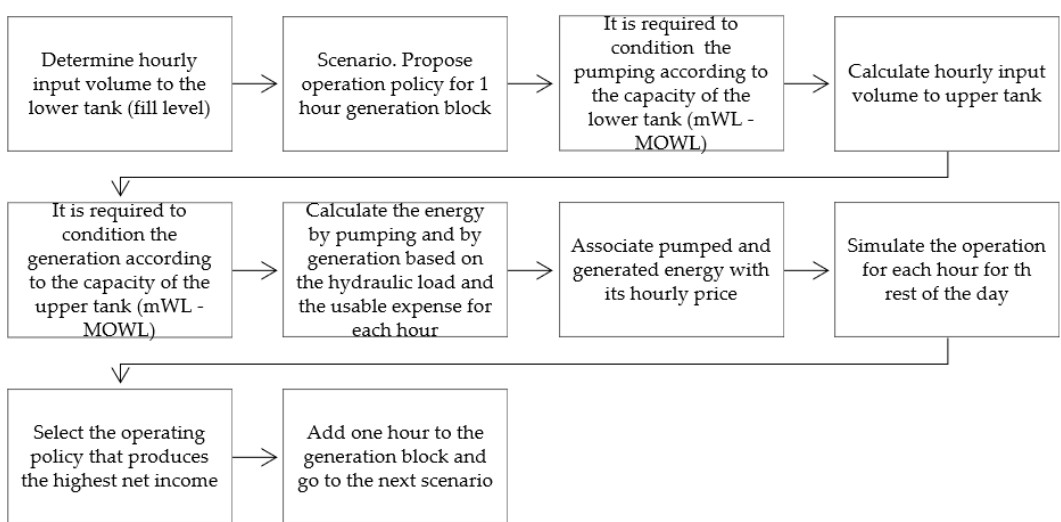

**Figure 10.** Optimization methodology for the operation policy.

The simulation tool presented here aims to assimilate the variations in demand and seasonal hydrological behaviors. To perform this, first, the operation of the reservoirs is modeled, where it is assumed that the hydrological effects of the basin are already

considered in the inflows to the reservoir. In the same way, the demand is modeled by using local marginal prices (PML), where seasonal variations are already considered.

Then, the model couples the behavior of the two variables, hydrological and demand, through the PMLs, optimizing the water resources based on the prices presented at each moment. Finally, the expected result is to obtain an hourly model that reports the energy dispatch as the best annual daily operation policy for a PHES.

### 2.4.1. Reservoir Simulation

The developed model simulates the evolution of the levels in the reservoir, being the core part of the hydroenergetic analysis. It is fundamentally used to select the optimal power and the characteristic levels of a power plant or dam [32]. For this case, it was allowed to determine the levels of the upper and lower regulation hydraulic tanks, see Tables 3 and 4.

With these values, the power and generation that the project would provide are determined, as well as the parameters that allow the sizing of the electromechanical equipment for the generation work [33].

To simulate the operation of the reservoir, the continuity equation (principle of mass conservation) is used, for a certain time interval, in this case, hourly:

$$VI - VO = \Delta V \tag{1}$$

where:

$VI$: Total inflow volumes during a time interval (hm$^3$)

$VO$: Total outflow volumes during the same time interval (hm$^3$)

$\Delta V$: Variation of the stored volume of the reservoir in the selected time interval (hm$^3$)

For the case study, there is the limitation that all the outflow volume due to PHES, either by turbine or spill, must be reincorporated into the reservoir of the hydroelectric facility to restore its usable volume at the end of the day.

Taking these elements into account, the continuity equation coupled to the hydraulic regulation tanks that make up the PHES is expressed as:

For the lower tank:

$$Ent_{LT} - Vol_{Pump} - Spill_{LT} = \Delta V_{LT} \tag{2}$$

where:

$Ent_{LT}$: Inflow volume to the Lower Tank, by own basin, (hm$^3$)

$Vol_{Pump}$: Pumped volume to the upper tank (hm$^3$)

$Spill_{LT}$: Spilled volume in the lower tank (hm$^3$)

$\Delta V_{LT}$: Variation of the volume stored in the lower tank (hm$^3$)

For the top tank:

$$Ent_{Pump} - Evap_{UT} - Vol_{Turb} - Spill_{UT} = \Delta V_{UT} \tag{3}$$

where:

$Ent_{Pump}$: Pumped volume from the lower tank (hm$^3$)

$Evap_{UT}$: Evaporation volume in the upper tank, (hm$^3$)

$Vol_{Turb}$: Turbined volume of the upper tank towards the reservoir, (hm$^3$)

$Spill_{UT}$: Spilled volume in the upper tank, (hm$^3$)

$\Delta V_{UT}$: Variation of the volume stored in the upper tank, (hm$^3$)

In the case of spills, it is considered that any excess volume that occurs with respect to the MOWL of any of the tanks (lower or upper) must be returned to the reservoir.

Therefore, with the turbine volume in the upper tank and the spills from both tanks returned to the hydroelectric reservoir, its operation is not altered.

It is important to clarify that this simulation restricts the pumping operation of the lower tank as soon as it fills up to its minimum water level, thus guaranteeing the required submergence. The calculation of the minimum submergence required to activate the pumping avoids the effect of cavitation in the mechanical equipment, according to the expression of the net positive suction height [34].

The same happens with the operation of the turbine in the upper tank; it begins its operation until filling is ensured at its wML.

Considering the previous sizing of the PHES (Tables 1–4), the energy consumed to power the pumping equipment is estimated, as well as the electrical energy produced at the hourly level and their respective powers (MW) at the hourly level.

For this, and according to the hydraulic loads (levels) that occur every hour, the expressions of energy power are used. Pump power is defined as:

$$Pow_{Pump} = \frac{\rho * g * Qb * Hb}{\eta} \tag{4}$$

where:

$Pow_{Pump}$: Power of the pumping equipment, (MW)
$\rho$: Density of water, (1000 kg/m$^3$)
$g$: Acceleration due to gravity, (9.81 m/s$^2$)
$Qb$: Flow rate pumped, (m$^3$/s)
$Hb$: Hydraulic head per pump, (m)
$\eta$: Hydraulic efficiency of electromechanical equipment, (85.64%)

Turbine power:

$$Pow_{Turb} = \rho * g * Qt * Ht * \eta \tag{5}$$

where:

$Pow_{Turb}$: Power of the turbine, (MW)
$\rho$: Density of water, (1000 kg/m$^3$)
$g$: Acceleration due to gravity, (9.81 m/s$^2$)
$Qt$: Flow rate pumped (m$^3$/s)
$Ht$: Hydraulic head per turbine, (m)
$\eta$: Hydraulic efficiency of electromechanical equipment, (85.64%)

In accordance with the fundamental energy equation in pressurized hydraulic systems, the local energy losses due to the conduit configuration are considered, either at the time of pumping or in the case of turbines, where a factor is considered for the geometry of the speed charge [35], dynamic energy losses are also taken into account with the help of the Darcy–Weisbach expressions and the Manning number [36]; with this, the friction on the walls of the conduit is established. Therefore, the following expression is used:

$$H_f = \frac{f * V^2 * L}{2g * D} \tag{6}$$

where:

$Hf$: Frictional head loss throughout the pipeline, (m)
$f$: Darcy friction losses (unitless)
$g$: Acceleration due to gravity, (9.81 m/s$^2$)
$D$: Diameter of pipe, (m)
$V^2/2g$: Velocity head, (m)
$L$: Length of the pipe, (m)

2.4.2. Optimization of the PHES Operation Policy

This optimization determines the benefits according to the hours it is pumped and the hours it is turbined to maximize the net income that is made up of the difference between selling and buying electricity according to the established scenarios.

The proposed objective function has the following form (see Equation (8)), where the sums go from $t = 1$ to $t = n$, and the terms are a function of t, which goes from hour 1 of 1 January 2018 to hour 23 of 31 December 2020.

$$FO = Max\left(\sum_{t_0=1}^{t=n} E_{turb} * PML_n - \sum_{t_0=1}^{t=n} E_{pump} * PML_n\right) \qquad (7)$$

where:

$FO$: Objective function of net income
$E_{turb}$: Turbined energy by the block of hours n, (MWh)
$E_{pump}$: Pumped energy by block of hours n, (MWh)
$PML_n$: Price of energy for each hour n, (USD/MWh)

The model that simulates the PHES operation considers leaving the generation time block permanent according to the proposed period, which ranges from 1 h to 9 h. To recirculate the same volume of water between the hydraulic regulation tanks, the pumping hours necessary for that same volume to feed the upper tank and thus complete the pumping-generation cycle are calculated. It is also necessary to restrict the minimum and maximum levels for the operation, as well as the condition of not-operating-simultaneously.

The above restrictions are expressed in Equations (8) to (11):

$$0 \leq time_{Pump} \leq 23\,\text{h} \qquad if\ Level_{LT} > mWL_{LT} \qquad (8)$$

$$time_{Pump} = 0,\ if\ Level_{UT} \geq MOWL_{UT} \quad or \quad LEvel_{LT} \leq mWL_{LT} \qquad (9)$$

$$0 \leq time_{Turbine} \leq 9\,\text{h} \quad if\ Level_{UT} > mWL_{UT} \qquad (10)$$

$$time_{Turbine} = 0,\ if\ Level_{UT} \leq mWL_{UT} \quad or \quad time_{Pump} \neq 0 \qquad (11)$$

where:

$time_{Pump}$: Pumping equipment operation, (hours)
$time_{Turbine}$: Turbine equipment operation, (hours)
$Level_{LT}$: Level of the lower tank, (masl)
$Level_{UT}$: Level of the upper tank, (masl)
$mWL_{LT}$: Minimum Water Level for lower tank, (masl)
$MOWL_{LT}$: Maximum Ordinary Water Level lower tank, (masl)
$mWL_{UT}$: Minimum Water Level for upper tank operation, (masl)
$MOWL_{UT}$: Maximum Ordinary Water Level upper tank, (masl)

According to the optimal schedules for the operation policy, the model reports the energy ratio indicator between generating and pumping electricity (G/B), represented by the following Equation (12), as well as the total spilled volume (Equation (13)).

Energy Ratio:

$$\frac{G}{B} = \frac{\sum_{t_0=1}^{t=n} E_{turb}}{\sum_{t_0=1}^{t=n} E_{pump}} \qquad (12)$$

where:

$E_{turb}$: Turbined energy by the block of hours n, (MWh)
$E_{pump}$: Pumped energy pumped by block of hours n, (MWh)

Spills:

$$Spillage = \sum_{t_0=1}^{t=n} Spill_{Ts} \qquad (13)$$

where:

$Spill_{Ts}$: Spilled volume of the upper tank in block n, (hm$^3$)

## 3. Results

The way to simulate scenario 1 is as follows: a schedule is proposed for the generation block of one hour, the hours required to pump the same volume to the upper tank are calculated (in this case, 1.71 h), then, for each generation block location alternative, covering between 11 a.m. and 11 p.m., all possible locations of the start of the pumping block between 0 a.m. and 10 a.m. are explored, in order to select the combination for the generation–pumping schedule, which leads to the maximum value of the objective function defined in Equation (8) (see Figure 11) in which the red block corresponds to the operation of the turbine, the blue block to the operation of the pump, and the green lines to the average prices of each month.

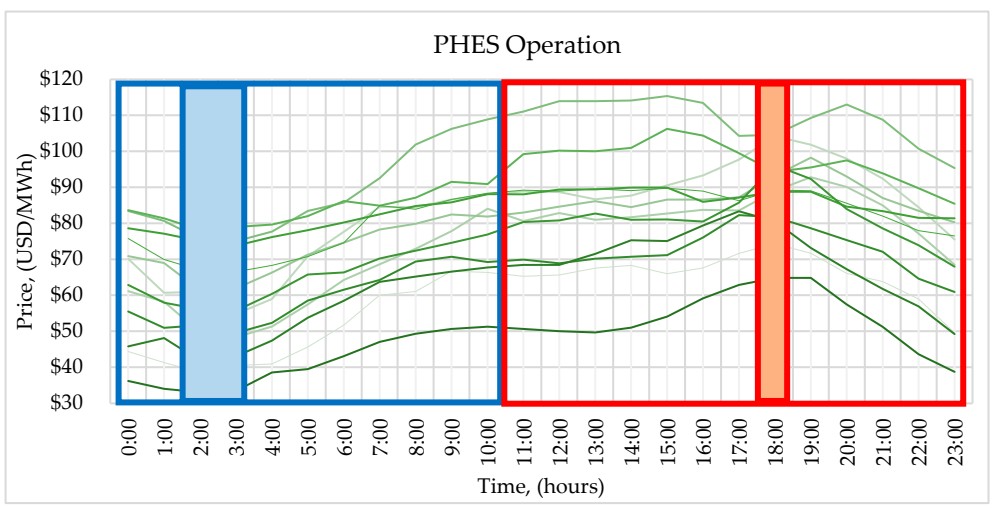

**Figure 11.** Representation of the optimization model. Own source with CENACE information, 2020.

Once the tour of the hours of the day of scenario 1 is finished, the simulation increases by another hour of generation, passing to the second scenario, 2 h in a row in the spawn block. The pumping hours are recalculated to obtain the volume to recirculate (3.09 h) and again sweep for the remaining hours of the day with these new operating blocks. The model then finishes the simulation to determine the optimal schedule for this scenario two.

This algorithm is repeated until considering a block of 9 h of generation, forming a total of 9 scenarios; this total of scenarios was determined because when the next step to a 10-h block is given, the pumping-generation cycle exceeds 24 h a day and to the limitation of not-operating-simultaneously (Equation (12)).

Table 8 shows the optimal schedule for each scenario, its generation block and its pumping block, including its production and consumption, as well as its plant factor. With these data, the energy ratio shown in percentage is determined (generation/pumping). In the last column, the average volume in a year is reported; the aim is to spill as little as possible since the water resource is the "fuel" of the PHES, so it is necessary not to lose more volume.

**Table 8.** Optimized function of reservoir operation.

| Generation (hrs) | Optimal Schedule (h-min) | Generation (GWh/Year) | Plant Factor | Pumping (hours) | Optimal Schedule (h-min) | Consumption (GWh/Year) | Plant Factor | Energy Ratio G/B | Discharge (hm³/Year) |
|---|---|---|---|---|---|---|---|---|---|
| | | | | | Reservoir Function Operation | | | | |
| 1.00 | 18:00–19:00 | 27.50 | 3.67% | 1.71 | 02:00–03:42 | 39.05 | 5.41% | 70.43% | 0.21 |
| 2.00 | 18:00–20:00 | 54.18 | 7.24% | 3.09 | 01:00–04:05 | 75.92 | 10.43% | 71.36% | 3.87 |
| 3.00 | 17:00–20:00 | 80.66 | 10.75% | 4.47 | 01:00–05:28 | 112.55 | 15.42% | 71.66% | 0.02 |
| 4.00 | 16:00–20:00 | 107.00 | 14.26% | 5.85 | 00:00–05:51 | 149.06 | 20.38% | 71.78% | 1.61 |
| 5.00 | 16:00–21:00 | 133.17 | 17.75% | 7.24 | 23:00–06:14 | 185.38 | 25.29% | 71.84% | 3.67 |
| 6.00 | 15:00–21:00 | 159.26 | 21.32% | 8.62 | 23:00–07:37 | 221.60 | 30.23% | 71.87% | 0.01 |
| 7.00 | 14:00–21:00 | 185.14 | 24.79% | 10.01 | 22:00–08:01 | 257.56 | 35.30% | 71.88% | 3.45 |
| **8.00** | **13:00–21:00** | **210.83** | **28.35%** | **11.39** | **22:00–09:23** | **293.16** | **40.08%** | **71.92%** | **0.03** |
| 9.00 | 12:00–21:00 | 236.88 | 31.78% | 12.77 | 22:00–10:46 | 329.03 | 44.89% | 71.99% | 0.66 |

Table 9 shows the results of the operation by energy arbitration service as the annual average of the simulation, and it is worth mentioning that the tool developed here is associated with the energy price recorded at the hourly level. Therefore, by maximizing the operating model based on historical hourly prices, the highest value is guaranteed for the generation block and the lowest prices for pumping blocks.

**Table 9.** Power Arbitrage Optimized Operation.

| Generation (hrs) | Optimal Schedule (h-min) | Sale (USD/MWh-y) | Purchase (USD/MWh-y) | Net Income (USD-year) | (%) |
|---|---|---|---|---|---|
| | | Energy Arbitration Results | | | |
| 1.00 | 18:00–19:00 | $2,357,631 | $1,613,055 | $744,575 | 146.2% |
| 2.00 | 18:00–20:00 | $4,598,865 | $3,160,280 | $1,438,584 | 145.5% |
| 3.00 | 17:00–20:00 | $6,766,164 | $4,754,404 | $2,011,760 | 142.3% |
| 4.00 | 16:00–20:00 | $8,872,570 | $6,402,297 | $2,470,272 | 138.6% |
| 5.00 | 16:00–21:00 | $10,966,154 | $8,148,441 | $2,817,714 | 134.6% |
| 6.00 | 15:00–21:00 | $13,053,673 | $9,978,717 | $3,074,956 | 130.8% |
| 7.00 | 14:00–21:00 | $15,092,054 | $11,860,094 | $3,231,959 | 127.3% |
| **8.00** | **13:00–21:00** | **$17,092,463** | **$13,837,750** | **$3,254,713** | **123.5%** |
| 9.00 | 12:00–21:00 | $19,103,383 | $15,908,307 | $3,195,077 | 120.1% |

According to Figure 12, where the G/B energy ratio is plotted against the annual net income, the optimal point for the proposed operation policy is identified.

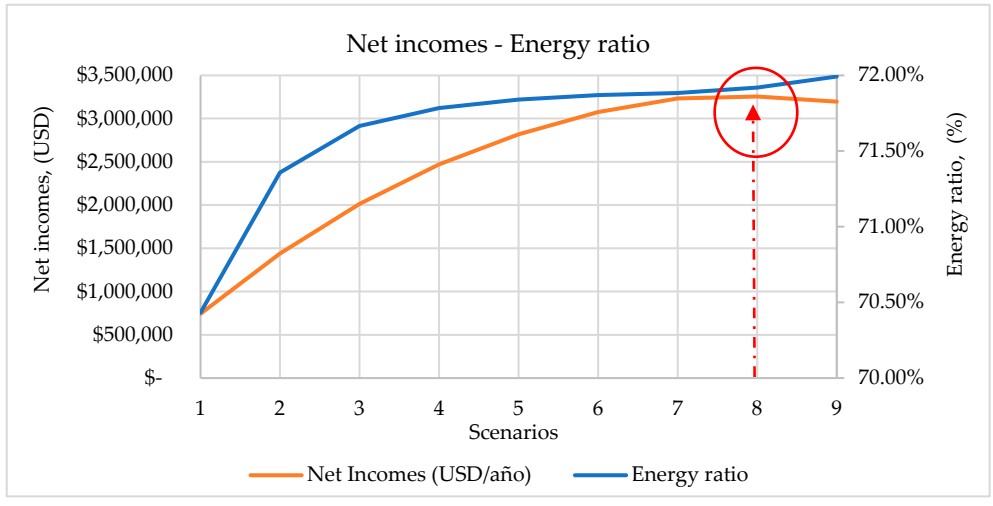

**Figure 12.** Comparison between net revenue and G/B energy ratio. Own source with analysis of results.

This way, scenario 8 (energy block of 8 h for electricity generation) offers the point of greatest benefit for net income since, toward scenario 9, the net income is diminished, see Table 9. On the other hand, the energy ratio presents an asymptotic function at approximately 72%, see Table 8.

Therefore, the operating policy is defined as follows:

- The optimum time for generation at the PHES Zimapán would be from 1:00 p.m. to 9.00 p.m. (8 continuous hours), producing 210 GWh/year with revenues of more than USD 17.1 million annually.
- The pumping schedule is from 22:00 to 09:23 (11:39 continuous hours), consuming 293 GWh/year and purchasing electricity for USD 13.8 million annually.
- In this way, the energy ratio is 71.92%, while the maximum net income contributes a value of more than USD 3.25 million (a 123.52% ratio with respect to the purchase of energy).
- The annual discharge would be 0.03 hm$^3$/year.

As a better reference, the behavior of the operation of the lower tank for the 3 years in question is plotted in Figure 13, where it is shown that the input volume is compatible with the existing hydroelectric installation. Therefore, in the dry season, the tank becomes completely empty (records below the mWL); the same situation happens for extreme events where the lower tank has to spill to maintain its level in the MOWL.

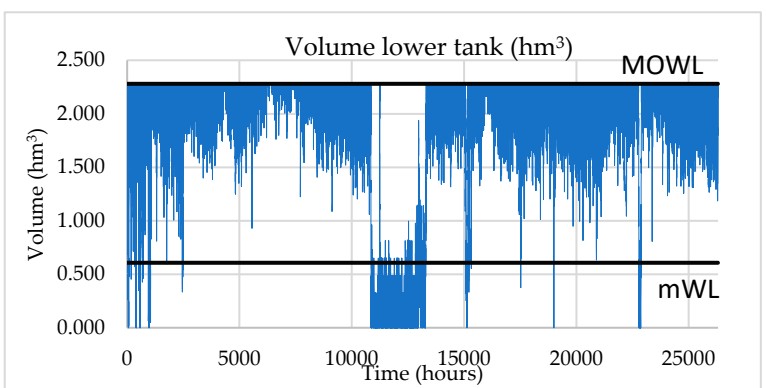

**Figure 13.** Behavior of the volume in the lower tank throughout the simulation, 3 years. Own source with analysis of results.

Figure 14 shows the behavior of the upper tank, where, due to the restrictions in its operation, it remains above the mWL and thus guarantees the operation works in turbine mode. For the cases where the tank is full, with more volume input, it is forced to spill. Figure 15 shows the volumes spilled for the 3-year period.

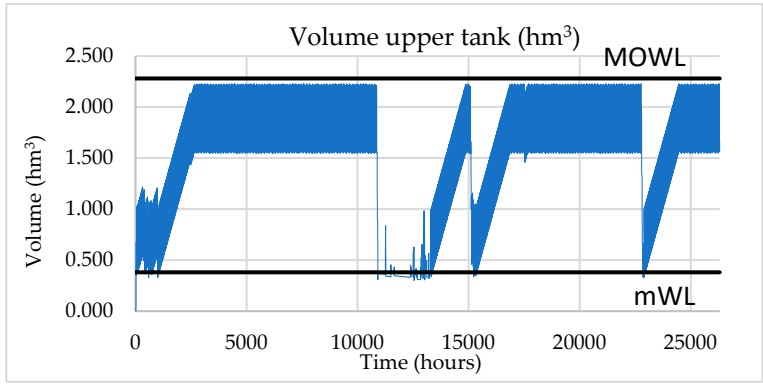

**Figure 14.** Behavior of the volume in the upper tank throughout the simulation, 3 years. Own source with analysis of results.

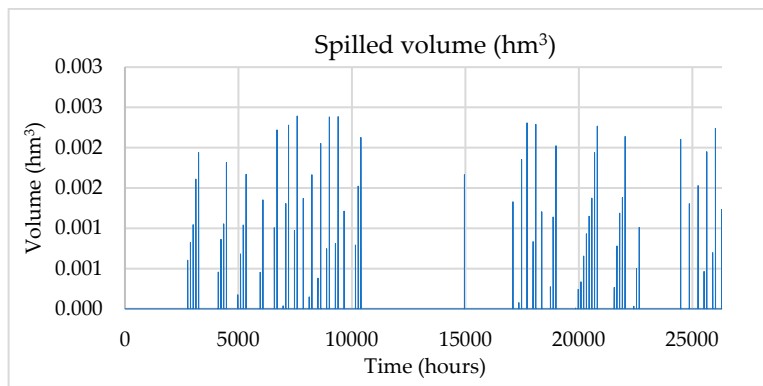

**Figure 15.** Behavior of the spilled volume in the upper tank throughout the simulation, 3 years. Own source with analysis of results.

To determine if these net incomes are sufficient to achieve viability, a preliminary exercise was carried out where the PHES Zimapán proposed here was submitted to an economic and financial evaluation. Table 10 shows the technical and economic parameters of the PHES Zimapán.

The financial parameters considered an evaluation horizon of 50 years, a discount rate of 10% with 50% debt, and an interest rate of 9.5% with a debt term of 25 years. The following results were obtained, shown in Table 11.

From the financial run, it turns out that the local marginal prices of the current electricity market do not have a differential wide enough to achieve profitability.

Subsequently, a sensitivity analysis is carried out for each economic-financial parameter, where two ways were found to achieve its viability.

**Table 10.** PHES Zimapán economic parameters.

| PHES Zimapán Technical Proposal, Economic Parameters | | | |
|---|---|---|---|
| Capacity to install | MW | 80 | Proposal |
| Total investment amount | USD | 140,000,000 | Investment unit cost is taken from Bulk Energy Storage: Economic Analysis (EPRI, 2016) |
| **Income** | | | |
| Total benefits | USD | 32,269,874 | Considering only the net income |
| **Costs** | | | |
| Total cost | USD | 141,979,935 | Made up of the payment of capital, operation, and maintenance, as well as the payment of debt |

**Table 11.** Results of the economic evaluation.

| Results | |
|---|---|
| Net Present Value | −USD109,710,61 |
| Capital IRR | −1.63% |
| Benefit/cost ratio | 0.23 |

## 4. Discussion

The model that simulates the operation of the PHES Zimapán basins at the hourly level considers hydraulic criteria and restrictions for its design and operation, as well as losses due to evaporation in the upper regulation hydraulic tank and energy losses due to hydraulic load due to conduction, both for the pumping mode and the turbine mode.

Derived from the height or hydraulic load that occurs at each moment and according to the determination of the stored volumes, the power and energy required to pump and produce electricity through the PHES are calculated.

This simulator considers the operation based on the mWL for both hydraulic tanks, always taking care of their submergence value. Likewise, when the level exceeds the MOWL, the excess volume is discharged and must return to the reservoir of the existing dam, so that the water resource is available for the PHES operation through its regulation tanks.

The optimum result for the PHES Zimapán, with a proposed capacity of 80 MW and a drop of 420 m to store about 2.26 cubic hectometers, obtains greater benefits by operating for 8 continuous hours in the high demand range (energy production 210.83 GWh /year) and pumping for almost 11.39 h in times of low demand (energy consumption 293.16 GWh/year). With this configuration, the PHES consumes 82.33 GWh/year more energy than it produces, see Table 8.

Additionally, when considering the service for energy arbitrage during the optimization of scenarios, a net income per year of more than USD 3.25 million was found for selling about USD 17.1 million per year and buying USD 13.8 million of energy in critical hours; this income is around 123.52% with respect to the annual energy purchase (Table 9).

The following Tables 12 and 13 show the summary of the results obtained for the PHES Zimapán case study with the optimal operation policy according to the energy arbitration service.

**Table 12.** Results of the reservoir function operation.

| Reservoir Function Operation | | | | | | |
|---|---|---|---|---|---|---|
| **PHES** | **Block of Hours** | **Optimal Schedule** | **Energy (GWh)** | **Plant Factor** | **Energy Ratio G/B** | **Discharge (hm³)** |
| Generation (hours) | 8 | 1:00 p.m.–9:00 p.m. | 210 | 28.35% | 71.92% | 0.03 |
| Pumping (hours) | 11.39 | 10:00 p.m.–9:23 a.m. | 293 | 40.08% | | |

**Table 13.** Results to the service by energy arbitration.

| Energy Arbitrage | | | | | | |
|---|---|---|---|---|---|---|
| **PHES** | **Block of Hours** | **Optimal Schedule** | **Sale (USD)** | **Purchase (USD)** | **Net Income (USD)** | **Income Ratio** |
| Generation (hours) | 8 | 1:00 p.m.–9:00 p.m. | 17,092,463 | 0.00 | 3,254,713 | 123.52% |
| Pumping (hours) | 11.39 | 10:00 p.m.–9:23 a.m. | 0.00 | 13,837,750 | | |

Figure 16 intends to show the hours in which the pumping-generation cycle is activated, associated with the average price of the energy, where, for hours of low demand, the water would be pumped (blue), while, in the hours of high demand, energy would be delivered through the turbines (yellow); trying to make the most of the hourly behavior of the demand.

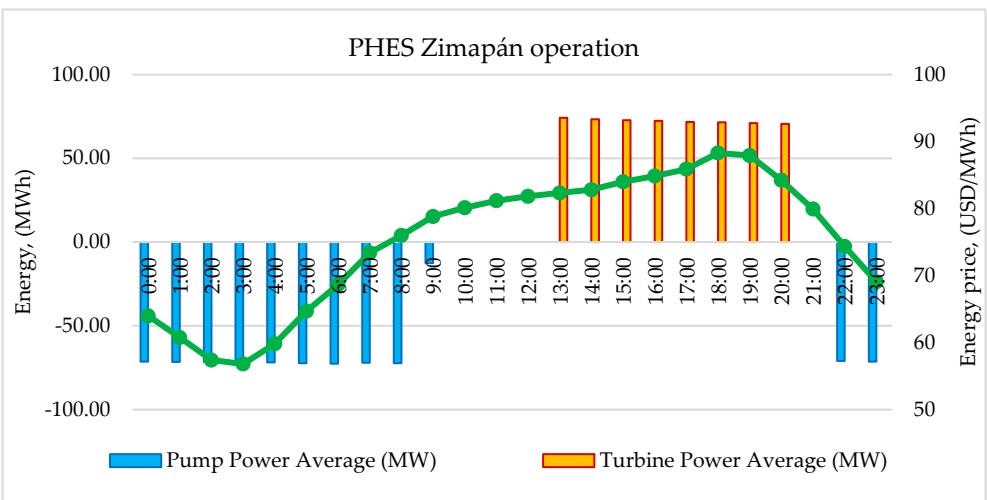

**Figure 16.** Optimum operation of the booster station. Own source with analysis of results.

In this sense, by using PHES within the electrical system, low-demand energy could be consumed and generated at times of high demand or when energy has a better economic value.

## 5. Conclusions

The growth in the supply of renewable energy has been mostly with intermittent plants whose generation regime does not satisfy the demands, which causes an oversupply and saturation of the electrical network in the hours of lower demand and causing a deficit in the offer, mainly in the hours of greatest demand, so it is essential to have energy stored on a large scale and the ability to use it immediately to cover this imbalance.

Until now, the hydroelectric plants have been responsible for covering the imbalance; however, as shown in Figure 1, this has forced them to work with a system for which they were not designed, concentrating their operation in a few hours, which causes efficiency problems and reduces their useful life.

Under these conditions, it is technically attractive to explore the use of PHES that can absorb the excessive supply of surplus energy in the hours of less demand and thus alleviate saturation problems, while this energy is stored for later delivery in the hours of greatest demand.

Given that the pumping–generation cycles of storage plants by pumping involve energy losses in the pipes, it is vitally important to optimize their operation by selecting the hours in which it is pumped and generated in such a way as to obtain maximum benefit by taking advantage of the differential of prices defined by the arbitration of the National Energy Control Center.

In this context, this work manages to simulate the behavior of a PHES located within a hydroelectric facility, the simulation tool proposed here determines the operating policy that maximizes the annual net profit (purchase–sale) and minimizes spills in the upper hydraulic tank taking as a case study the preliminary proposal of the PHES Zimapán.

Since the CFE shared public data on the Zimapán reservoir, the simulation model was applied to this specific hydroelectric power station.

With the simulation tool presented here, the operation policy of a PHES is obtained based on its water resources and the energy prices presented in the node to which it is interconnected.

In this way, this optimization could be replicated in those hydroelectric power station candidates to place PHES in their reservoirs; that was the purpose of sharing this document with a broad audience.

The optimization of the operating policy for the PHES Zimapán, with a proposed capacity of 80 MW and a drop of 420 m with the capacity to store some 2.26 cubic hectometers

of water, obtains greater benefits by generating energy for 8 continuous hours in the high demand range, producing 210.83 GWh/year, while pumping operates for almost 11.39 h at times of low demand, reaching an energy consumption of 293.16 GWh/year. With this configuration, the PHES consumes 82.33 GWh/year more energy than it produces.

It is recognized that, although the energy consumed by pumping is greater than the energy generated by the turbine, this technology yields economic benefits when using the energy arbitration service.

With this approach, a profit is obtained by a net income of just over USD 3.25 million a year, by selling around USD 17.1 million a year and buying USD 13.8 million of energy in critical hours; this income ratio is around 123.52% with respect to the annual purchase of energy.

Finally, this annual net income value was subjected to a financial economic evaluation in order to determine the feasibility of the PHES, finding as a result that, at this time, the price differential is not wide enough to be able to implement this type of project in Mexico using only the arbitration service.

**Author Contributions:** Conceptualization, G.A.; methodology, G.A., M.L.A. and R.D.; formal analysis, G.A.; research, G.A.; writing—preparation of the original draft, G.A. and R.D.; writing—review and editing, M.L.A., O.F. and R.D. All authors have read and agreed to the published version of the manuscript.

**Funding:** This research was funded by CONACYT, grant number 000219—doctorate in civil engineering.

**Institutional Review Board Statement:** Not applicable.

**Informed Consent Statement:** Not applicable.

**Data Availability Statement:** Not applicable.

**Acknowledgments:** The authors wish to thank the Federal Electricity Commission (CFE) for the data provided from official sources and the UNAM Institute of Engineering for its technical support.

**Conflicts of Interest:** The authors declare no conflict of interest.

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
