# Peer review of "Optimal Schedule the Operation Policy of a Pumped Energy Storage Plant Case Study Zimapán, México"

_electronics, doi:10.3390/electronics11244139_

Round 1

Reviewer 1 Report

The paper is well written and the case study taken as Mexico is well appreciated.

I have only few corrections as suggestions.

In conclusion, the SDG 7 influence can be written.

For instance, refer the works of author Rishi Pugazhendhi and cite few papers in introduction as it's essential to talk about the impact of SDG 7 and also about RE assessment. 

Is it possible to draw a conceptual diagram explaining the overall concept at the end of introduction?

Reviewer 2 Report

Dear Authors,

The paper is quite interesting, especially on the costs. A tabular cost-benefit analysis would have significantly improved the results section and presented a clear picture of the income. Maybe this can be looked into as a suggestion. Also, the simulation tool used is not mentioned in the paper. It would be worthwhile by the authors to consider mentioning what tools and algorithms were applied for simulation.

Reviewer 3 Report

Very well written and detailed introduction, lays out the background, and the research question. However the literature review part can be made a little bit more explicit. This will allow the reader to clearly figure out what has been published already and how this study addresses any gap.

In methods, we do need some initial clarity on why this specific PHES facility was chosen. 

For prices (Table 7), given that there's clearly a seasonal component to prices - I think a range is useful, clearly defining when in a given year prices are supposed to be highest or lowest for any hour. 

For evaporation data (Table 6) - how exactly was the transfer between 1993-1995 and 2018-2020 done? Do we just assume the same values, or are they scaled based on some sort of a national/regional average?

I am assuming reservoir simulation and optimization was done with the aid of some software. Details of that should be included, with appropriate citations and a brief overview of settings/configurations as applicable.

I think Figure 15 is a good way to summarize the results. However a little bit more detail can be given in order to make it clear that what does the results mean to regulators or utility planners. I think a seasonal component should also be brought in to show how can the optimum values vary over a given year - given that electricity demand and supply from other sources will also vary as well.

If doing seasonal analysis adds significant burden on the researchers, then this limitation should be clearly explained. However, this will limit what conclusions you can draw, and how useful these conclusions are to examples beyond this specific PHES. This is the case right now. Given that the paper doesn't say why this PHES was specifically chosen, and the conclusion doesn't mention any external relevance of the results - then as a reader, I am wondering what is the purpose of sharing this paper with a broad audience.

Reviewer 4 Report

You may also include emergency demand in case of certain unexpected situations like earthquakes, hurricanes etc. Also you may address fault diagnosis in energy distribution. This paper deals with pumped-storage hydroelectric plants. An optimal operation policy is proposed. They used a simulation tool for  knowing annual benefits under the energy arbitration service. They took the case of Zimapan hydroelectric facility at hourly level for a 3 years. They show that energy consumption for a given rate of generation is optimal. 

Round 2

Reviewer 3 Report

The revisions have sufficiently addressed my comments.